# Uptake of COVID-19 vaccine among high-risk urban populations in Southern Thailand using the COM-B model

**Charuai Suwanbamrung**[1,2], **Benchawan Srinam**[2], **Pakawan Promkool**[2], **Warissara Suwannakarn**[2], **Sangchom Siripanich**[1,2], **Md. Siddikur Rahman**[3], **Muhammad Haroon Stanikzai**[4] *

**1** Excellent Center for Dengue and Community Public Health (EC for DACH), Walailak University, Nakhon Si Thammarat, Thailand, **2** Public Health Research Program, School of Public Health, Walailak University, Nakhon Si Thammarat, Thailand, **3** Department of Statistics, Begum Rokeya University, Rangpur, Bangladesh, **4** Department of Public Health, Faculty of Medicine, Kandahar University, Kandahar, Afghanistan

* haroonstanikzai1@gmail.com

**Data Availability Statement:** The authors confirm that all data underlying the findings are fully available without restriction. All relevant data are

## Abstract

### Background

The COVID-19 pandemic has imposed unprecedented suffering on social and individual levels worldwide. Vaccines against COVID-19 have been prioritized as a crucial strategy for ending the pandemic as well as minimizing its consequences.

### Objectives

This study aimed to determine the uptake of COVID-19 vaccine among high-risk urban populations in Southern Thailand using the Capability, Opportunity, Motivation, and Behavior (COM-B) model.

### Methods

We conducted a web-based cross-sectional study in the Hat Yai district, Songkhla province in Southern Thailand, in September and October 2021. The questionnaire was composed of sections on sociodemographic characteristics, COVID-19 vaccination status, and COM-B constructs. We employed a multivariable logistic regression analysis to determine factors associated with the uptake of the COVID-19 vaccine. We set statistical significance at *p < 0.05*.

### Results

In this study, females constituted 54.7% of the total participants (n = 358), and nearly half of the participants (45.8%) were in the younger age group (18–29). Of all the participants, 59.5% (95%CI: 54.2%-64.6%) received at least one dose of the COVID-19 vaccine. Factors associated with the uptake of COVID-19 vaccine and their adjusted OR (95% CI) were being married: 3.59 (2.06–6.24), having a graduate degree: 2.34 (1.38–3.96), gainfully

within the paper and its Supporting Information Files.

**Funding:** This study was financially supported by the Excellent Center for Dengue and Community Public Health [WU-COE-66-16], School of Public Health, Walailak University. The funders had no role in study design, data collection and analysis, decision to publish, or preparation of the manuscript.

**Competing interests:** The authors have declared that no competing interests exist.

employed: 3.30 (1.91–5.67), having a high level of opportunity: 2.90 (1.48–5.66), and having a high level of motivation: 2.87 (1.17–17.08).

## Conclusion

The uptake of COVID-19 vaccines was moderate in this population. Moreover, the results showed that the COM-B model is useful in predicting COVID-19 vaccine uptake. The findings of this study could be used to aid future public health interventions in any event of outbreaks similar to COVID-19 disease in Thailand and beyond.

## Introduction

Over the past three years, the COVID-19 pandemic has resulted in a growing burden of biopsychosocial problems in developing and developed countries [1, 2]. On an individual level, the disaster has resulted in an overwhelming number of bio-psycho-social deficits in the general population that impacted the physical, psychological, and social aspects of their health and well-being [3, 4]. In response to this largest health crisis of our time, the development and deployment of effective vaccines has been prioritized as a crucial strategy to contain the spread of the virus and reduce its impact on public health [5, 6].

Following the catastrophic first wave of the COVID-19 pandemic, multi-national pharmaceutical industries made vaccines available sporadically during the second wave and in large amounts after the second wave [6]. The administration of COVID-19 vaccines has dramatically shifted the nature of the disease, leading to a marked reduction in the number of cases and deaths [7, 8]. Immediately following the successful development of multiple effective and safe COVID-19 vaccines, the World Health Organization (WHO) urged all countries to vaccinate at least 70% of their populations by mid-2022, with priority given to vaccinating health workers and the most vulnerable groups (e.g., individuals over 60 years of age, those with compromised immune system, and those with pre-existing medical conditions) [9].

However, the proportion of people vaccinated against COVID-19 reflects significant global disparities. Moreover, vaccination rates are suboptimal, particularly in the developing countries. Data assert that approximately 70.3% of the world's population has received at least one dose of the COVID-19 vaccine [10]. In developing countries, however, this proportion is only 32.3% [11]. In Thailand, vaccination escalated rapidly, where about 82.5% of the population has received at least their first dose of the COVID-19 vaccine as of June 2023 [11], with notably lower coverage in remote areas [12].

Although vaccination remains a cornerstone of the COVID-19 pandemic response, broad public support remains elusive. For instance, a recent global systematic review found that the global pooled acceptance rate of the COVID-19 vaccine was 64.9%, with significant variations across WHO regions (range; 60.8%-81.6%) [13]. Other relevant studies have also found that the acceptance rate of the COVID-19 vaccine varies substantially from country to country and even from region to region within a country [14, 15]. In Thailand, the acceptance rates of the COVID-19 vaccine reported ranged from 58% to 95.6% [12, 13, 16]. However, the above studies were mainly among vulnerable populations and emanated from regions other than the South.

Consistent findings of extensive research indicate that the decision to vaccinate against COVID-19 is influenced by diverse factors, including sociodemographic characteristics, sociocultural and religious considerations, political perspectives, trust in healthcare professionals

and current vaccines, the availability and accessibility of vaccination services, and fear of COVID-19 [12, 14, 17–19]. Studies from Thailand reported a strong association of sociodemographics and health system influencers with the COVID-19 vaccination decision [12, 15].

There is a growing recognition that theory-based behavioral models can effectively predict COVID-19 preventive behaviors [20, 21]. The capability, opportunity, motivation, and behavior (COM-B) model is recognized as an efficacious framework for preventing COVID-19 disease and acts as a practical framework for designing and promoting preventive behaviors [21, 22]. The results of several studies indicate that the application of this model is successful in COVID-19 prevention [23, 24]. There is little knowledge in the study area of the COM-B model, predicting the uptake of COVID-19 vaccines.

This study conducted in 2021 captured a period when the uptake of COVID-19 vaccines was suboptimal worldwide. However, at the time of writing this paper, the situation for COVID-19 vaccine acceptance has improved significantly. Recognizing the challenges encountered at the initial stages of COVID-19 vaccine acceptance, it becomes even more critical to identify gaps in the adoption of healthy behavior to devise effective interventions and implementation approaches. As an example, the COM-B model in COVID-19 vaccine uptake can provide implicit insights from behavior change models as a potential solution to any event of outbreaks similar to COVID-19 disease in Thailand and beyond.

## Materials and methods

### Study settings and design

A web-based cross-sectional study was conducted in the Hat Yai district, an urban area of Songkhla Province in Southern Thailand, in September and October 2021. The district is divided into sixteen administrative units or sub-districts and is lodging approximately 286,274 people [25]. We selected the Hat Yai district for the present study due to its designation as one of the areas significantly affected by the COVID-19 pandemic. Moreover, this district, characterized by certain demographics, cultural factors, or socioeconomic conditions could provide broader implications for public health policies and interventions in other developing regions of the world.

### Study population

We recruited a total of 358 participants who were currently active members of the Facebook group HATY AIZ (a social media platform dedicated to community engagement and connection) to participate in an anonymous survey using the online Google survey forms via Facebook Messenger. Our participants mainly lived in cities, were young, spoke Thai, and were willing to participate in this study.

### Sample size and sampling procedures

The estimated sample size was 325, obtained by a sample size calculation from the Facebook group HATY AIZ records, which documented the number of active adult members (800 members; July–August 2021) with a 95% confidence level, a 5% margin of error, a design effect of 1.5, and a 10% nonresponse rate. We increased the calculated sample size from a minimum of 325 participants to 379 participants in the prevision of missing data. The final analyses consist of 358 participants with their complete data sets. We used convenience sampling (voluntary participation) to recruit our participants in this study.

## Study measures

Based on relevant literature, we developed a structured questionnaire with sections on sociodemographic information, COM-B constructs, and participants' intentions to get vaccinated against COVID-19.

In this study, we used six sociodemographic characteristics as independent variables: sex (male/female); age (18–29, 30–44, 45–59, >60); marital status (currently married, currently unmarried); education (undergraduate, graduate or higher); occupation (occupation with uncertain income, occupation with regular income); and income in Baht ($\leq 26000$, $> 26000$).

We employed the comprehensive theoretical model grounded in the COM-B model to understand the factors influencing the individuals' behavior toward receiving the COVID-19 vaccination [20, 21]. The adapted COM-B model has three constructs: (1) capability, (2) opportunity, and (3) motivation with good psychometric properties for facilitating behavioral change.

The capability and opportunity constructs were measured using 12-item scales: capability (two negatives and ten positives) and opportunity (three negatives and nine positives). Each item was scored from 0 (unfavorable response) to 1 (favorable response), which yields a total score from 0 to 12. In each scale, a score of $\geq 10$ was used to signify a high level of capability and opportunity for receiving the COVID-19 vaccination, as defined in prior work [20, 26].

We employed a 5-point Likert scale to measure the motivation construct. The motivation construct consists of 12 items, including 6-item positive and 6-item negative subscales. Each item in the positive constructs was scored from 5 (strongly agree) to 0 (strongly disagree) and vice versa in the negative constructs, yielding a total score from 12 to 60. A score of $\geq 48$ was used to signify a high level of motivation for receiving the COVID-19 vaccination [26, 27].

The outcome variable of this study was the respondents' behavior regarding the uptake of the COVID-19 vaccine. The participants in this survey were asked about their COVID-19 vaccination status and the responses were in a dichotomous 'yes' or 'no' format.

## Data collection

The questionnaire, which consisted of sections on socio-demographic information, COVID-19 vaccine-related information, and the COM-B model, was initially drafted in English and later translated into Thai (local language) for the ease of administration. Before the commencement of the study, we pretested the questionnaire in another Facebook group with 65 participants to check and revise (if required) its verbal consistency. Additionally, we checked the structure reliability of the questionnaire in the pretested sample, and the Cronbach's Alpha value (Thai version) for capability, opportunity, motivation, and total was 0.86, 0.77, 0.82, and 0.83, respectively.

We distributed the Thai version of the questionnaire through the HATY AIZ Facebook group and were available for responses from September 10, 2021, to October 31, 2021. Once a potential client accessed the online survey form, a consent form appeared on the first page indicating the study description, objectives, and participants' right to withdraw at any time. If the client would like to participate, they could willfully provide their consent and choose their desired language of the form. Only then did the clients complete and submit the anonymous survey questionnaire.

## Statistical analysis

We employed descriptive statistics to understand participants' capability, opportunity, and motivation to adopt the new behavior, i.e., acceptance of the COVID-19 vaccine. We employed multivariable logistic regression analysis to determine factors associated with the uptake of the

COVID-19 vaccine. In all analyses, the assumptions for the multivariable logistic regression model were met. We set statistical significance at $p < 0.05$.

## Ethical consideration

The Ethics Committee for Human Research, Walailak University approved this study (ref. no. WUEC-21-214-01; Dated: August 16, 2021). All participants agreed to the terms of an electronic consent form before they could participate in the study. The electronic consent form included information on study description, objectives, and participants' right to withdraw at any time. Moreover, we followed the ethical principles outlined in the Declaration of Helsinki.

## Results

Table 1 indicates the sociodemographic characteristics of our study participants. A total of 358 Facebook users from the HATY AIZ group aged 18–60 years were included in this study. In our sample, 54.7% (196) were female, and nearly half of the participants (45.8%, 164) were young (18–29). Approximately one-third (42.7%, 153) of our participants were married, and

**Table 1. Sociodemographic characteristics of the study participants (n = 358).**

| Variables | Frequency (%) |
|---|---|
| **Age (In completed years)** | |
| 18–29 | 164 (45.8) |
| 30–44 | 104 (29.1) |
| 45–59 | 80 (22.3) |
| $\geq 60$ | 10 (2.8) |
| **Sex** | |
| Male | 162 (45.3) |
| Female | 196 (54.7) |
| **Marital status** | |
| Single | 192 (53.6) |
| Married | 153 (42.7) |
| Separated/Divorced | 13 (3.6) |
| **Educational status** | |
| Secondary school (level 1–3) | 9 (2.5) |
| Secondary school (level 4–6) | 41 (11.4) |
| Vocational Certificate/Diploma | 72 (20.1) |
| Bachelor degree | 220 (61.5) |
| Higher education | 16 (4.5) |
| **Employment status** | |
| Employed with regular income | 231 (64.5) |
| Employed with uncertain income | 127 (35.5) |
| **Monthly household income (in Baht)** | |
| $\leq 26000$ | 197 (55.0) |
| $> 26000$ | 161 (45.0) |
| **History of travel to/or from another province in the last 14 days** | |
| Yes | 35 (9.8) |
| No | 323 (90.2) |
| **History of exposure to COVID-19 patients in the last 14 days** | |
| Yes | 29 (8.1) |
| No | 329 (91.9) |

more than two-thirds (64.5%, 231) were gainfully employed. The monthly household income was ≤ 26000 Baht (750 USD, July 2023) in more than half of the participants (55%, 197). Data on participants' educational attainment and travel history are summarized in Table 1.

## Capability to receive COVID-19 vaccine

The mean score of the capability construct was 10.80 (± 1.35 SD) with a range of 3–12 points, and nearly two-thirds (249; 69.6%) of participants showed a high level of capability. Table 2 illustrates the detailed reflections of our participants on the capability construct within the COM-B model.

## Opportunity to receive COVID-19 vaccine

The mean score of the opportunity construct was 10.04 (± 1.72 SD) with a range of 2–12 points, and about 41.9% (150) of participants had a high level of opportunity to receive the COVID-19 vaccine. The detailed opportunity construct and participants' reflections are illustrated in Table 3.

## Motivation to receive COVID-19 vaccine

The mean score of the motivation construct was 41.18 (± 8.81 SD) with a range of 24–60 points, and a small number of the participants (12%, 43) had a high level of motivation to receive the COVID-19 vaccine. The detailed motivation construct and participants' reflections are illustrated in Table 4.

At the time of this study, around two-thirds (59.5%, 95%CI: 54.2%-64.6%) of study participants had received at least one dose of the COVID-19 vaccine.

**Table 2. Capability to receive COVID-19 vaccine (n = 358).**

| Items | Reponses, Frequency (%) | |
|---|---|---|
| | **Yes** | **No** |
| Knowledge of the spread of COVID-19 disease | 351 (98) | 7 (2) |
| Knowledge of the signs and symptoms associated with COVID-19 disease | 347 (96.9) | 11 (3.1) |
| COVID-19 vaccine can boost immunity | 329 (91.9) | 29 (8.1) |
| Knowledge of the vaccine doses and schedule | 264 (73.7) | 94 (26.3) |
| It is necessary to get enough sleep and refrain from consuming alcohol, tea, and coffee before vaccination | 344 (96.1) | 14 (3.9) |
| Individuals can go home without any surveillance and are not required to wait for 30 minutes to observe any symptoms post vaccination* | 75 (20.9) | 283 (79.1) |
| Knowledge of factors that increase the risk for severe COVID-19 | 344 (96.1) | 14 (3.9) |
| COVID-19 can be transmitted easily within families when a family member has signs and symptoms of COVID-19 disease | 346 (96.6) | 12 (3.4) |
| COVID-19 doesn't affect individuals' daily activities* | 82 (22.9) | 276 (77.1) |
| COVID-19 results in unemployment and income loss | 346 (96.6) | 12 (3.4) |
| Vaccination can be reserved online or through public channels | 335 (93.6) | 23 (6.4) |
| Decisions to receive the vaccine can be affected by its adverse effects | 302 (84.4) | 56 (15.6) |
| | **High** | **Low** |
| **Level of capability (cut-off point 90%)** | 249 (69.6) | 109 (30.4) |

* Negative capability

**Table 3. Opportunity to receive COVID-19 vaccine (n = 358).**

| Items | Reponses, Frequency (%) | |
|---|---|---|
| | **Yes** | **No** |
| You are supported by family and friends to receive the vaccine | 324 (90.4) | 34 (9.5) |
| You are informed to register for the vaccine | 303 (84.6) | 55 (15.4) |
| You live in a high-risk area for the COVID pandemic | 313 (87.4) | 45 (12.6) |
| You think that shopping malls and companies are sources of COVID-19 infection | 326 (91.1) | 32 (8.9) |
| You need a vaccine due to the disease's impact on your employment* | 320 (89.4) | 38 (10.6) |
| You need the government to provide vaccines for all citizens | 333 (93.0) | 25 (7.0) |
| You ask for a vaccine that fulfills all your needs* | 228 (63.7) | 130 (36.3) |
| Vaccine is available near your residence | 293 (81.8) | 65 (18.2) |
| The risk of infection is not increased if individuals are not vaccinated* | 155 (43.7) | 203 (56.7) |
| Adherence to social distancing measures lowers your risk of the COVID-19 disease | 316 (88.3) | 42 (11.7) |
| You can select the type of vaccine | 238 (66.5) | 120 (33.5) |
| You can travel to receive vaccine | 342 (95.5) | 16 (4.5) |
| | **High** | **Low** |
| **Level of opportunity (cut-off point 90%)** | 150 (41.9) | 208 (58.1) |

* Negative opportunity

## Factors associated with the uptake of COVID-19 vaccine in our sample

A multivariable logistic regression analysis indicated that being married (AOR = 3.59, 95%CI: 2.06–6.24), having a graduate degree (AOR = 2.34, 95%CI: 1.38–3.96), employed with a regular income (AOR = 3.30, 95%CI: 1.91–5.67), having a high level of opportunity (AOR = 2.90, 95% CI: 1.48–5.66), and having a high level of motivation (AOR = 2.87, 95%CI: 1.17–17.08) were associated with uptake of the COVID-19 vaccine (Table 5).

**Table 4. Motivation to receive COVID-19 vaccine (n = 358).**

| Items | Responses, Frequency (%) | | | | |
|---|---|---|---|---|---|
| | **Strongly agree** | **Agree** | **Neutral** | **Disagree** | **Strongly disagree** |
| You have fears about the spread of COVID-19 infection* | 102 (28.5) | 45 (12.6) | 58 (16.2) | 57 (17.9) | 96 (26.8) |
| You are worried that you might be infected with COVID-19 in the future* | 133 (37.2) | 44 (12.3) | 75 (20.9) | 51 (14.2) | 55 (15.4) |
| COVID-19 vaccines have side effects* | 1 (0.3) | 6 (1.7) | 48 (13.4) | 79 (22.1) | 224 (62.6) |
| You are afraid of the COVID-19 vaccine side effects* | 118 (33.0) | 35 (9.8) | 77 (21.5) | 55 (15.4) | 73 (20.4) |
| COVID-19 vaccines are necessary for the control of the pandemic | 281 (78.5) | 51 (14.2) | 24 (6.7) | 0 (0.0) | 2 (0.6) |
| COVID-19 vaccines prevent infection rather than severity* | 78 (21.8) | 38 (10.6) | 67 (18.7) | 52 (14.5) | 123 (34.4) |
| Individuals receive vaccines to avoid the COVID-19 infection | 163 (45.5) | 44 (12.3) | 47 (13.1) | 42 (11.7) | 62 (17.3) |
| Most individuals receive vaccines only to maintain their rights* | 106 (29.6) | 40 (11.2) | 71 (19.8) | 48 (13.4) | 93 (26.0) |
| Have trust in COVID-19 vaccines provided by the government | 66 (18.4) | 62 (17.3) | 74 (20.7) | 36 (10.1) | 120 (33.5) |
| Have trust in COVID-19 vaccines as they are registered by FDA | 100 (27.9) | 74 (20.7) | 78 (21.8) | 30 (8.4) | 76 (21.2) |
| Information about COVID-19 vaccines affects the decision to receive the vaccine | 243 (67.9) | 66 (18.4) | 44 (12.3) | 2 (0.6) | 3 (0.8) |
| You trust healthcare professionals' advice to receive the COVID-19 vaccine | 236 (65.9) | 68 (19.0) | 36 (10.1) | 4 (1.1) | 14 (3.9) |
| | | | | **High** | **Low** |
| **Level of motivation (cut-off point 90%)** | | | | 43 (12.0) | 315 (88.0) |

* Negative motivation

**Table 5. Factors associated with the uptake of COVID-19 vaccine; crude and adjusted odds ratio with 95% CI.**

| Independent Variables | Categories | Vaccination status | | Crude Odds Ratio (95% CI) | p-value | Adjusted Odds Ratio (95% CI) | p-value |
|---|---|---|---|---|---|---|---|
| | | Yes | No | | | | |
| Age | 18–44 | 147 | 121 | 1 | 0.01 | - | - |
| | ≥45 | 66 | 24 | 2.26 (1.33–3.82) | | | |
| Sex | Male | 99 | 63 | 1.05 (0.87–1.27) | 0.20 | - | - |
| | Female | 114 | 82 | 1 | | | |
| Marital status | Currently married | 115 | 38 | 3.30 (2.09–5.22) | 0.03 | 3.59 (2.06–6.24) | 0.02 |
| | Currently unmarried | 98 | 107 | 1 | | 1 | |
| Education status | Undergraduate | 61 | 61 | 1 | 0.01 | 1 | 0.03 |
| | Graduate | 152 | 84 | 1.81 (1.16–2.82) | | 2.34 (1.38–3.96) | |
| Employment status | Uncertain income | 114 | 117 | 1 | <0.001 | 1 | <0.001 |
| | Regular income | 99 | 28 | 3.62 (2.21–5.93) | | 3.30 (1.91–5.67) | |
| Level of capability | Low | 190 | 125 | 1 | 0.21 | - | - |
| | High | 23 | 20 | 0.75 (0.39–1.43) | | | |
| Level of opportunity | Low | 167 | 125 | 1 | <0.001 | 1 | <0.001 |
| | High | 46 | 20 | 1.72 (0.97–3.05) | | 2.90 (1.48–5.66) | |
| Level of motivation | Low | 177 | 138 | 1 | <0.001 | 1 | <0.001 |
| | High | 36 | 7 | 4.01 (1.73–9.28) | | 2.87 (1.17–7.08) | |

## Discussion

This study discovered the association between the COM-B constructs and COVID-19 vaccination behaviors. To the best of our knowledge, few documents have been published about the COVID-19 vaccine uptake via the COM-B model constructs. The study is important, as often time is not taken to understand how theory-based behavioral change models influence people's behavior toward COVID-19 vaccinations and what are the needs for effective immunization programs from a population perspective. This is informative as the population's behaviors toward an effective immunization program may differ from region to region, with significant implications for policymaking and restructuring immunization services.

In this study, we found that 59.5% of our study participants had received at least one dose of the COVID-19 vaccine. The results of previous studies demonstrated that the uptake of the COVID-19 vaccine in the Thai population ranged from 58% to 95.6% [12, 13, 16]. Also, national statistics reported that the uptake of the COVID-19 vaccine in the general population of Thailand was 82.5% [11]. Therefore, the COVID-19 vaccination rate in this study is lower than that recorded in most previous studies and national statistics. This difference may be attributed to a small sample size for the population sourced in this study. The possibility of differences in sociodemographics, study design, and geographic locations cannot be excluded. The acceptance and uptake of vaccines are affected by multiple factors including geographic location, time, sociodemographics, politics, culture, and vaccine type [13, 28]. Hence, we recommend further studies involving larger and geographically diverse populations to verify these findings.

Our study showed that the education level of the participants was significantly associated with the uptake of the COVID-19 vaccine. This finding is similar to earlier pertinent literature that pointed out that higher educational attainment is strongly associated with the acceptance and uptake of the COVID-19 vaccine [13, 28, 29]. Surprisingly, a similar study in Iran found an inverse relationship, which is contrary to our findings [30]. The decrease in acceptance of the COVID-19 vaccine among educated Iranians may coincide with the initiation of

vaccination in Iran before the findings of large clinical trials were announced. However, studies have shown that higher education levels may have a significant effect on immunization coverage [13, 28]. Given the findings of this study, providing credible information through various mediums coupled with the enhancement of health literacy through educational interventions and community messages could be useful in adopting a new health behavior, particularly for those with lower educational attainment. Moreover, these interventions could also be effective in addressing any remaining COVID-19 vaccine hesitancy in the community.

Consistent with relevant literature, our findings indicate that COVID-19 vaccine uptake was lower in currently unmarried participants than those currently married. Studies have found that vaccine uptake was higher in participants who were married [28, 31]. Given these findings, vaccination policies and strategies specific to populations with single/separated marital status for future pandemics are crucial, particularly considering specific concerns or preferences in such cases.

We noted that employed participants with a regular income had 3.3 times higher odds of getting vaccinated than those with unreliable income. Similar studies have shown an association between employment status and COVID-19 vaccination status [28, 30]. Employment vaccination requirements may have played some role as the data were collected after vaccination mandates by the Thai government. In fact, stable income is a crucial socioeconomic factor influencing people's health attitudes and behaviors. Therefore, considering socioeconomic factors such as income and employment is crucial when addressing the health needs of the population.

It is well known that theory-based behavioral models are crucial in adopting a new behavior [32–34]. We observed that 69.6%, 41.9%, and 12% of our study participants, respectively, had a high level of capability, opportunity, and motivation to receive the COVID-19 vaccine according to the COM-B model. In addition, we found that participants with high levels of opportunity and motivation were more likely to receive their first dose of the COVID-19 vaccine than participants with low levels of opportunity and motivation. Several studies reported that factors that influence behavioral intentions toward the COVID-19 vaccine are multi-dimensional [28, 29]. A study in Thailand found that the constructs in the COM-B model could successfully generate themes of behaviors that affect COVID-19 prevention behaviors [23]. Other studies have revealed that the adapted COM-B model provides refined details for each construct in the vaccine context [22, 24]. Based on the COM-B constructs, effective interventions could be designed based on the predictors of capability, opportunity, and motivation which can be beneficial in a variety of disaster, geographic and socioeconomic contexts to improve healthy behaviors.

## Limitations

Our findings have several limitations. The smaller sample size allows less precision in the findings. Considering the use of an online survey approach, elder participants and participants with low education may be under-represented in our sample. We employed the Thai version of the COM-B model in the context of the COVID-19 vaccine for the first time, while their psychometric properties require assertion. We have not assessed the bio-psycho-social health of our subjects, and their socio-cultural affiliations that may have confounded their uttered responses. Finally, the findings of this study are limited to a particular period of the pandemic, and since then there have been substantial improvements in the acceptance of the COVID-19 vaccine.

## Conclusion

We found that 59.5% of our study participants had received at least one dose of a COVID-19 vaccine. The uptake of the COVID-19 vaccine was influenced by the sociodemographics of

our participants (i.e., educational attainment, marital status, and employment status). We also found that opportunity and motivation constructs of the COM-B model effectively predicted the uptake of the COVID-19 vaccines. The findings of this study could be used to inform evidence-based interventions in a variety of disaster, geographic and socioeconomic contexts to improve healthy behaviors.

## Supporting information

**S1 Dataset. Microsoft excel file with minimal dataset.**
(XLSX)

## Acknowledgments

The authors sincerely thank all the people in the high-risk district who were involved in the study for their assistance and support.

## Author Contributions

**Conceptualization:** Charuai Suwanbamrung, Benchawan Srinam, Warissara Suwannakarn, Sangchom Siripanich, Md. Siddikur Rahman, Muhammad Haroon Stanikzai.

**Data curation:** Muhammad Haroon Stanikzai.

**Formal analysis:** Pakawan Promkool, Warissara Suwannakarn, Muhammad Haroon Stanikzai.

**Funding acquisition:** Charuai Suwanbamrung, Pakawan Promkool.

**Investigation:** Warissara Suwannakarn, Muhammad Haroon Stanikzai.

**Methodology:** Benchawan Srinam, Pakawan Promkool, Warissara Suwannakarn, Md. Siddikur Rahman, Muhammad Haroon Stanikzai.

**Project administration:** Pakawan Promkool, Muhammad Haroon Stanikzai.

**Resources:** Pakawan Promkool.

**Software:** Muhammad Haroon Stanikzai.

**Supervision:** Charuai Suwanbamrung, Benchawan Srinam, Sangchom Siripanich, Md. Siddikur Rahman, Muhammad Haroon Stanikzai.

**Validation:** Warissara Suwannakarn, Muhammad Haroon Stanikzai.

**Visualization:** Muhammad Haroon Stanikzai.

**Writing – original draft:** Charuai Suwanbamrung, Benchawan Srinam, Warissara Suwannakarn, Sangchom Siripanich, Md. Siddikur Rahman, Muhammad Haroon Stanikzai.

**Writing – review & editing:** Charuai Suwanbamrung, Benchawan Srinam, Warissara Suwannakarn, Sangchom Siripanich, Md. Siddikur Rahman, Muhammad Haroon Stanikzai.

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
