## [Decision Letter · Decision Letter 0]

2 Nov 2023

PONE-D-23-22650Uptake of COVID-19 vaccine among high-risk urban populations in Southern Thailand using the COM-B ModelPLOS ONE

Dear Dr. Stanikzai,

Thank you for submitting your manuscript to PLOS ONE. After careful consideration, we feel that it has merit but does not fully meet PLOS ONE’s publication criteria as it currently stands. Therefore, we invite you to submit a revised version of the manuscript that addresses the points raised during the review process.

We look forward to receiving your revised manuscript.

Kind regards,

Kittisak Jermsittiparsert, Ph.D.

Academic Editor

PLOS ONE

6. We note that Figure 1 in your submission contain [map/satellite] images which may be copyrighted. All PLOS content is published under the Creative Commons Attribution License (CC BY 4.0), which means that the manuscript, images, and Supporting Information files will be freely available online, and any third party is permitted to access, download, copy, distribute, and use these materials in any way, even commercially, with proper attribution. For these reasons, we cannot publish previously copyrighted maps or satellite images created using proprietary data, such as Google software (Google Maps, Street View, and Earth). For more information, see our copyright guidelines: http://journals.plos.org/plosone/s/licenses-and-copyright.

Additional Editor Comments:

Overall, this manuscript was written quite well and received very positive feedback from reviewers. If this manuscript is submitted for consideration at the end of 2021 or early 2022, it will need to be accepted and published quickly as it is a study to face the urgent situation at that time. However, now that the COVID-19 situation has passed and interest in vaccination has decreased a lot because it is not something that people still don't know even a little bit about it that might be helpful. Looking at the details of this manuscript, I found that these are issues that many studies have already addressed. This is especially true when considering the population of this study that is not unique or may provide new, deeper or more interesting answers. To make this manuscript interesting and useful for publication at this time, the authors need to improve this manuscript in at least two parts.

1) Point out how important and interesting the southern region of Thailand is within a context of a developing country. However, there should also be strong empirical data to support it.

2) If it is not possible to change the independent variables used in the study, the authors should not only point out what variables are present but also explain the interesting points from the discovery of the relationship between those variables. This is not only to answer questions about COVID-19, but also to deal with future outbreaks in this type of area.

This means that the authors need to be more explicit about the issues being studied and how they are more important or interesting than in many previous studies. Moreover, it is a discussion and expansion to show how the results of this research can be further developed to obtain answers that are more useful than existing research.

Reviewers' comments:

Reviewer's Responses to Questions

**Comments to the Author**

1. Is the manuscript technically sound, and do the data support the conclusions?

Reviewer #1: Yes

Reviewer #2: Yes

2. Has the statistical analysis been performed appropriately and rigorously? 

Reviewer #1: Yes

Reviewer #2: Yes

3. Have the authors made all data underlying the findings in their manuscript fully available?

Reviewer #1: Yes

Reviewer #2: Yes

4. Is the manuscript presented in an intelligible fashion and written in standard English?

Reviewer #1: Yes

Reviewer #2: Yes

5. Review Comments to the Author

Reviewer #1: I have no concerns about the issue of republishing. This research is a new research study in the context of the southern region of Thailand. and have research ethics standards certified by the host university In addition, this research also writes useful research results that will lead to future public health guidelines in Thailand and countries with similar contexts.

Reviewer #2: The study is ethically sound, receiving approval from Walailak University's Ethics Committee and obtaining informed consent from participants. The research demonstrates exceptional originality, as confirmed by a favorable plagiarism assessment. It appropriately references current literature, covering a wide range of topics related to COVID-19 and vaccination. The study design, a web-based cross-sectional approach in a high-risk urban area, aligns with its objectives and uses academically sound measures. Statistical analysis and interpretation are generally appropriate, though presenting confidence intervals for percentages and means and providing more detailed interpretation of logistic regression results could enhance clarity. Overall, the conclusions are well-supported by data and analyses, highlighting a 59.5% vaccine uptake rate and identifying demographic factors and the influence of the COM-B model on vaccine acceptance. The study suggests further research and provides valuable insights for evidence-based interventions. For the full details of my reviews, please refer to the attached MS WORD file, named "SRIRATH-SENT1112023.docx."

6. PLOS authors have the option to publish the peer review history of their article (what does this mean?). If published, this will include your full peer review and any attached files.

Reviewer #1: **Yes: **Naksit Sakdapat

Reviewer #2: **Yes: **Srirath Gohwong, Ph.D.

---

## [Author Response · Author response to Decision Letter 0]

21 Dec 2023

Dear Editor,

We would like to thank the editor and the reviewers for their thoughtful evaluation of our manuscript entitled “Uptake of COVID-19 vaccine among high-risk urban populations in Southern Thailand using the COM-B Model”. Please find our revised manuscript along with tracked changes file (yellow highlighted), which we believe is substantially strengthened now that we have incorporated reviewers’ recommendations. 

We made amendments according to the journal requirements.

Additional Editor Comments:

Overall, this manuscript was written quite well and received very positive feedback from reviewers. If this manuscript is submitted for consideration at the end of 2021 or early 2022, it will need to be accepted and published quickly as it is a study to face the urgent situation at that time. However, now that the COVID-19 situation has passed and interest in vaccination has decreased a lot because it is not something that people still don't know even a little bit about it that might be helpful. Looking at the details of this manuscript, I found that these are issues that many studies have already addressed. This is especially true when considering the population of this study that is not unique or may provide new, deeper or more interesting answers. 

Response: Thank you so much for your positive comments. We would like to add the findings of this study could be of significant importance in adopting a new behavior through application of the COM-B model in any given disaster. We have provided more details for the practical implications of our key findings (yellow highlighted in the manuscript).

To make this manuscript interesting and useful for publication at this time, the authors need to improve this manuscript in at least two parts.

1. Point out how important and interesting the southern region of Thailand is within a context of a developing country. However, there should also be strong empirical data to support it.

Response: Thank you so much. We agree with your comment and added details in the introductions section and methodology, and pointed out this point through out the discussion of our revised manuscript (Please see lines 95-102, lines 106-112, please see full discussion). 

2. If it is not possible to change the independent variables used in the study, the authors should not only point out what variables are present but also explain the interesting points from the discovery of the relationship between those variables. This is not only to answer questions about COVID-19, but also to deal with future outbreaks in this type of area. This means that the authors need to be more explicit about the issues being studied and how they are more important or interesting than in many previous studies. Moreover, it is a discussion and expansion to show how the results of this research can be further developed to obtain answers that are more useful than existing research.

Response: Thank you so much. We are indeed very thankful for this feedback. We have revised most sections of our manuscript and addressed this important issue of how our findings can be important in future events like COVID-19 disease. Moreover, we have expanded our discussion of how our findings can be useful (Please see lines 338-340, lines 343-345, lines 364-369, lines 372-375, lines 380-383, lines 394-397, and lines 411-415). We have also highlighted key findings for policy consideration. Thank you once again and we really hope that our manuscript would be acceptable to you.

Response to Reviewers’ comments

Response to Reviewer 1 comments

I have no concerns about the issue of republishing. This research is a new research study in the context of the southern region of Thailand, and as research ethics standards certified by the host university. In addition, this research also writes useful research results that will lead to future public health guidelines in Thailand and countries with similar contexts.

Response: Thank you so much for recommending the publication of our manuscript. 

Response to Reviewer 2 comments

1. The study is ethically sound, receiving approval from Walailak University's Ethics Committee and obtaining informed consent from participants. The research demonstrates exceptional originality, as confirmed by a favorable plagiarism assessment. It appropriately references current literature, covering a wide range of topics related to COVID-19 and vaccination. The study design, a web-based cross-sectional approach in a high-risk urban area, aligns with its objectives and uses academically sound measures. 

Response: Thank you so much for constructive and positive feedback.

2. Statistical analysis and interpretation are generally appropriate, though presenting confidence intervals for percentages and means and providing more detailed interpretation of logistic regression results could enhance clarity. 

Response: Thank you so much for this feedback. We have added 95% CI for COVID-19 vaccines percentages, added means for COM-B constructs, and added p-value to the logistic regression results (Please see the results).

3. Overall, the conclusions are well supported by data and analyses, highlighting a 59.5% vaccine uptake rate and identifying demographic factors and the influence of the COM-B model on vaccine acceptance. The study suggests further research and provides valuable insights for evidence-based interventions.

Response: Thank you so much.

Responses to reviewer 2 detailed feedback.

1. The research complies with ethical standards for experimentation and research integrity, as it was granted approval by Walailak University's Ethics Committee (reference number WUEC-21-214-01, dated August 16, 2021). Informed consent was diligently secured from all participants through an official online consent form, as provided by the committee. This form transparently communicated the study's goals, objectives, and the participants' entitlement to withdraw at any point. Furthermore, the study steadfastly adhered to the ethical principles outlined in the Declaration of Helsinki, which stands as one of its outstanding strengths.

Response: Thank you so much.

2. The title of this study, "Enhancing COVID-19 Vaccine Uptake in High-Risk Urban Populations in Southern Thailand through the Application of the COM-B Model," boldly showcases its unique and innovative approach. The assessment for potential plagiarism conducted by Chulalongkorn University resulted in a favorable outcome, revealing a similarity index of 0.00% (Source: [http://plag.grad.chula.ac.th], [http://www.chula.ac.th/en/]). This academic research demonstrates exceptional originality and quality, and the plagiarism assessment findings reinforce my confidence in its academic integrity.

Response: Thank you so much for positive feedback.

3. Is the work clearly and accurately presented and does it cite the current literature?

The work does cite a range of current literature relevant to the research topic. The references included in the provided text encompass a variety of sources, including academic journals, international organizations like the World Health Organization (WHO), and primary research studies. The references cover a wide range of topics related to the COVID-19 pandemic, vaccination, and factors influencing vaccine acceptance.

Response: Thank you so much.

4. Is the study design appropriate and does the work have academic merit? Are Experiments, statistics, and other analyses performed to a high technical standard and are described in sufficient detail?

a. The study design is well-suited to the research objectives, involving a web-based cross-sectional study conducted in Hat Yai district, an urban area in southern Thailand significantly affected by the COVID-19 pandemic. The study's focus on COVID-19 vaccine uptake among high-risk urban populations is justified. The measures, including socio-demographic characteristics and the adapted Capability, Opportunity, Motivation, and Behavior (COM-B) model, are academically sound. The data collection process is well documented, with pretesting and reliability checks. The statistical analysis using SPSS, descriptive statistics, and multivariable logistic regression aligns with the study's objectives. The use of a significance level (P < .05) is appropriately stated. 

Response: Thank you so much.

b. However, providing more detail on specific statistical tests, control variables, and effect sizes would enhance the analysis, especially multivariable logistic regression. Logistic multiple regression is a statistical approach employed when analyzing a binary outcome variable, such as yes/no or 1/0, in relation to multiple independent variables (predictors or covariates). The author should reveal the specific conditions and assumptions of Logistic multiple regression: the outcome must be binary, observations should be independent, a linear relationship between predictors and log-odds is assumed, multicollinearity should be avoided, a sufficient sample size is needed, outliers need attention, perfect separation can pose challenges, model fit must be evaluated, odds ratios should be appropriately interpreted, and causal relationships should be established through experimental designs or advanced statistical methods. Adhering to these considerations is vital for ensuring the robustness and reliability of results. Collaboration with a statistician or data analyst is recommended for addressing complex modeling issues and assessing potential assumption violations. 

Response: Thank you so much. We agree with your comments and have added the suggested details. In our methods, we have declared that all assumptions for multivariable logistic regression analysis were met (Please see line 177-178).

c. The limitations of the study are well-acknowledged, including the smaller sample size, potential under-representation of certain demographic groups, and the fact that the study focused on a particular period of the pandemic. The study's recognition of these limitations demonstrates a thorough understanding of the research's scope and potential constraints.

Response: Thank you so much for the positive feedback.

5. If applicable, are the statistical analysis and its interpretation appropriate?

a. The statistical analysis and interpretation of the data are generally appropriate. The study presents a comprehensive analysis of the socio-demographic characteristics of the participants, their capability, opportunity, motivation, and the factors associated with COVID-19 vaccine uptake. The use of descriptive statistics, such as percentages and means, is suitable for presenting the data. The logistic regression analysis effectively identifies factors associated with the vaccine uptake. 

Response: Thank you so much.

b. However, there is room for minor improvements. While the results are well presented, it would be beneficial to include the confidence intervals for the percentages and means to indicate the precision of the estimates. Moreover, the interpretation of the logistic regression results could be more detailed, explaining the adjusted odds ratios and their implications more explicitly. In conclusion, the study's analysis and interpretation are sound, but enhancing clarity and precision in presenting results and their implications would further strengthen the paper.

Response: Thank you so much for the constructive feedback. We have added the suggested points to the results (please see the results section). Moreover, we have detailed the implications of our findings in the discussion section (Please see lines 338-340, lines 343-345, lines 364-369, lines 372-375, lines 380-383, lines 394-397, and lines 411-415). 

6. Are the conclusions drawn adequately supported by the results? [Conclusions are presented in an appropriate fashion and are supported by the data.]

The study's conclusions find robust support in the data and analyses. It reports a COVID-19 vaccine uptake rate of 59.5% among participants, backed by the data's clear portrayal of vaccination status and construct scores for capability, opportunity, and motivation. Furthermore, the study uncovers various demographic factors linked to vaccine acceptance, such as marital status, education, and employment, as indicated by the odds ratios derived from multivariable logistic regression analysis. Notably, the study underscores the effectiveness of the COM-B model's opportunity and motivation constructs in driving vaccine uptake, leveraging comprehensive data from Tables 2, 3, and 4. It wisely calls for future research encompassing larger and more diverse populations to validate these findings and offers guidance on evidence-based interventions for enhancing COVID-19 vaccine uptake based on the identified factors and COM-B constructs.

Response: Thank you so much for clarifying these important findings and providing positive feedback to our manuscript.

At last, we would like to thank respected editor and respected reviewers for the overall positive feedback on our manuscript and hope that it will be acceptable to them.

Sincerely,

---

## [Decision Letter · Decision Letter 1]

22 Feb 2024

PONE-D-23-22650R1Uptake of COVID-19 vaccine among high-risk urban populations in Southern Thailand using the COM-B ModelPLOS ONE

Dear Dr. Stanikzai,

Thank you for submitting your manuscript to PLOS ONE. After careful consideration, we feel that it has merit but does not fully meet PLOS ONE’s publication criteria as it currently stands. Therefore, we invite you to submit a revised version of the manuscript that addresses the points raised during the review process.

Dear author:

Thank you for the revision and taking the efforts to address all comments. To further improve the manuscript's readability, we would like to invite you to address the below.

1. Abstract" Add in the total number of sample size after the word the word Participant= Results: In this study, females constituted 54.7% of the total participants (**n=XX**)...

2. Given these findings, vaccination polices and strategies specific to populations with single/separated marital status for future pandemics are crucial, particularly considering the** likelihood of lower social support** in such cases. This is not clear in term of what do you meant by likelihood of lower social support and how is it related to the single/separated marital status?

3. Please format the tables according to PLOS ONE requirement.  

4. Please attached a letter of proofread of the entire revised manuscript.

5. Please help to provide at short 1 line on "HATY AIZ"- what is HATY AIZ?

6. In the procedure section, please state the right to withdraw as well as token of appreciation- if applicable.

7. Please change all p-value in upper case e.g., in here "statistical significance at **P < 0.05**" (line 178, abstract) to lower case and italic "*p <0.05*"

8. Seeking your elaboration on Capability, Opportunity, Motivation, and Behavior. In line 145, you mentioned that "In each scale, a score of >10 was used to signify a high level of capability' but in line 204 and 205, it was also stated that "The mean score of the capability construct was 9.53 (± 1.55 SD) with a range of 3-12 points, and 205 nearly two-thirds (249; 69.6%) of them showed a high level of capability". 9.53 certainly is not >10. Hence, I am not sure how did you arrive at the conclusion that "205 nearly two-thirds (249; 69.6%) of them showed a high level of capability". Same concern on "Opportunity to receive COVID-19 vaccine". Can you please clarify this for better readability? 

9. Please include this article in your in text citation - line 78 to support the claim on "trust in healthcare professionals" - https://bmcpublichealth.biomedcentral.com/articles/10.1186/s12889-022-12632-z  (citation requests are optional)

We look forward to receiving your revised manuscript.

Kind regards,

Pei Boon Ooi, Ph.D.

Academic Editor

PLOS ONE

Journal Requirements:

Additional Editor Comments:

Dear author:

Thank you for the revision and taking the efforts to address all comments. To further improve the manuscript's readability, we would like to invite you to address the below.

1. Abstract" Add in the total number of sample size after the word the word Participant= Results: In this study, females constituted 54.7% of the total participants (n=XX)...

2. Given these findings, vaccination polices and strategies specific to populations with single/separated marital status for future pandemics are crucial, particularly considering the likelihood of lower social support in such cases. This is not clear in term of what do you meant by likelihood of lower social support and how is it related to the single/separated marital status?

3. Please format the tables according to PLOS ONE requirement.

4. Please attached a letter of proofread of the entire revised manuscript.

5. Please help to provide at short 1 line on "HATY AIZ"- what is HATY AIZ?

6. In the procedure section, please state the right to withdraw as well as token of appreciation- if applicable.

7. Please change all p-value in upper case e.g., in here "statistical significance at P < 0.05" (line 178, abstract) to lower case and italic "p"

8. Seeking your elaboration on Capability, Opportunity, Motivation, and Behavior. In line 145, you mentioned that "In each scale, a score of >10 was used to signify a high level of capability' but in line 204 and 205, it was also stated that "The mean score of the capability construct was 9.53 (± 1.55 SD) with a range of 3-12 points, and 205 nearly two-thirds (249; 69.6%) of them showed a high level of capability". 9.53 certainly is not >10. Hence, I am not sure how did you arrive at the conclusion that "205 nearly two-thirds (249; 69.6%) of them showed a high level of capability". Same concern on "Opportunity to receive COVID-19 vaccine".

9. Please include this article in your in text citation - line 78 to support the claim on "trust in healthcare professionals" - https://bmcpublichealth.biomedcentral.com/articles/10.1186/s12889-022-12632-z

Reviewers' comments:

Reviewer's Responses to Questions

**Comments to the Author**

1. If the authors have adequately addressed your comments raised in a previous round of review and you feel that this manuscript is now acceptable for publication, you may indicate that here to bypass the “Comments to the Author” section, enter your conflict of interest statement in the “Confidential to Editor” section, and submit your "Accept" recommendation.

Reviewer #1: All comments have been addressed

Reviewer #2: All comments have been addressed

2. Is the manuscript technically sound, and do the data support the conclusions?

Reviewer #1: Yes

Reviewer #2: Yes

3. Has the statistical analysis been performed appropriately and rigorously? 

Reviewer #1: Yes

Reviewer #2: Yes

4. Have the authors made all data underlying the findings in their manuscript fully available?

Reviewer #1: Yes

Reviewer #2: Yes

5. Is the manuscript presented in an intelligible fashion and written in standard English?

Reviewer #1: Yes

Reviewer #2: Yes

6. Review Comments to the Author

Reviewer #1: It is a particularly interesting study that studies the use of COVID-19 vaccines. In groups High-risk urban populations in southern Thailand. The model is integrated opportunity, motivation, and behavior (COM-B Model). The findings are detailed comprehensive and beneficial to Thailand's public health in the future. It can be used to plan policy measures on intervention measures in the event of an outbreak similar to COVID-19.

Reviewer #2: The study, titled "Uptake of COVID-19 vaccine among high-risk urban populations in Southern Thailand using the COM-B Model," provided a clear and accurate presentation, utilizing a web-based cross-sectional approach in the Hat Yai district, Southern Thailand. It strengthened its findings by drawing on a diverse array of contemporary literature, encompassing socio-economic implications, global vaccine acceptance rates, and factors influencing vaccine acceptance. Through an extensive citation of current literature, the research established a robust contextual foundation, weaving insights from reviews on socio-economic impacts, paradigms of vaccine development, and studies on vaccine acceptance rates into its analytical framework. The comprehensive and up-to-date engagement with relevant sources enhanced the study's credibility, providing a well-supported backdrop for its research findings. "I am honored to have the opportunity to read this paper."

7. PLOS authors have the option to publish the peer review history of their article (what does this mean?). If published, this will include your full peer review and any attached files.

Reviewer #1: No

Reviewer #2: **Yes: **Srirath G., Ph.D.

---

## [Author Response · Author response to Decision Letter 1]

24 Feb 2024

Dear Editor,

We would like to thank the editor and the reviewers for their positive evaluation of our manuscript entitled “Uptake of COVID-19 vaccine among high-risk urban populations in Southern Thailand using the COM-B Model”. Please find our revised manuscript along with tracked changes file (yellow highlighted), which we believe is substantially strengthened now that we have incorporated editorial recommendations. 

We made amendments according to the journal requirements.

Additional Editor Comments:

Thank you for the revision and taking the efforts to address all comments. To further improve the manuscript's readability, we would like to invite you to address the below.

Response: We also would like to thank the respected editor and reviewers for their approval of our responses and revision. Each point was very helpful for making our paper better and suitable for publication in your prestigious journal. As such, we have tried to amend the manuscript in light of all your and reviewers’ comments and the journal guideline. Please find our responses to the editorial requests.

1. Abstract" Add in the total number of sample size after the word the word Participant= Results: In this study, females constituted 54.7% of the total participants (n=XX)...

Response: Thank you so much for you’re your constructive feedback. We have added the sample size (please see page: 2, line: 29).

2. Given these findings, vaccination polices and strategies specific to populations with single/separated marital status for future pandemics are crucial, particularly considering the likelihood of lower social support in such cases. This is not clear in term of what do you meant by likelihood of lower social support and how is it related to the single/separated marital status?

Response: Thank you. We agree with you that this section was not clear and revised as suggested (please see lines 295 and 296).

3. Please format the tables according to PLOS ONE requirement. 

Response: Thank you. We formatted our tables according to PLOS ONE requirement.

4. Please attached a letter of proofread of the entire revised manuscript.

Response: Thank you so much for this suggestion. All authors re-edited the manuscript to remove the grammatical errors and improve the manuscript. Besides, we have sent the manuscript to English language expert in US (our colleague) and she has proofread it. Please let us know if any error remains.

5. Please help to provide at short 1 line on "HATY AIZ"- what is HATY AIZ?

Response: Thank you so much. We agree with your suggestion and have added a line on HATY AIZ group (please see line 113).

6. In the procedure section, please state the right to withdraw as well as token of appreciation- if applicable.

Response: Thank you. In our research, participants had the right to withdraw at any stage and we have added these details into the procedure section (please see lines 163 and 185).

7. Please change all p-value in upper case e.g., in here "statistical significance at P < 0.05" (line 178, abstract) to lower case and italic "p <0.05"

Response: Thank you so much. We made changes as suggested (please see line 26, line 175, and Table 5 heading).

8. Seeking your elaboration on Capability, Opportunity, Motivation, and Behavior. In line 145, you mentioned that "In each scale, a score of >10 was used to signify a high level of capability' but in line 204 and 205, it was also stated that "The mean score of the capability construct was 9.53 (± 1.55 SD) with a range of 3-12 points, and 205 nearly two-thirds (249; 69.6%) of them showed a high level of capability". 9.53 certainly is not >10. Hence, I am not sure how did you arrive at the conclusion that "205 nearly two-thirds (249; 69.6%) of them showed a high level of capability". Same concern on "Opportunity to receive COVID-19 vaccine". Can you please clarify this for better readability? 

Response: Thank you so much for your meticulous observation. We checked our descriptive statistics again and found that there was some error in typing and the mean scores have been corrected now. We have also checked our other analyses and the findings were correctly typed in other cases, except this section which was highlighted by the editor. Additionally the cut-off score should have been written ≥ 10 (typo mistake). We have made these corrections too. Please see the changes on lines 143, 208, 217, and 231.

9. Please include this article in your in text citation - line 78 to support the claim on "trust in healthcare professionals" - https://bmcpublichealth.biomedcentral.com/articles/10.1186/s12889-022-12632-z (citation requests are optional)

Response: Thank you so much for this suggestion. We have added this relevant citation into our manuscript. 

Response to Reviewers’ comments

Response to Reviewer 1 comments

Comment: It is a particularly interesting study that studies the use of COVID-19 vaccines. In groups High-risk urban populations in southern Thailand. The model is integrated opportunity, motivation, and behavior (COM-B Model). The findings are detailed comprehensive and beneficial to Thailand's public health in the future. It can be used to plan policy measures on intervention measures in the event of an outbreak similar to COVID-19.

Response: Thank you so much for your constructive feedback and recommendation on the publication of our manuscript. 

Response to Reviewer 2 comments

Comment: The study, titled "Uptake of COVID-19 vaccine among high-risk urban populations in Southern Thailand using the COM-B Model," provided a clear and accurate presentation, utilizing a web-based cross-sectional approach in the Hat Yai district, Southern Thailand. It strengthened its findings by drawing on a diverse array of contemporary literature, encompassing socio-economic implications, global vaccine acceptance rates, and factors influencing vaccine acceptance. Through an extensive citation of current literature, the research established a robust contextual foundation, weaving insights from reviews on socio-economic impacts, paradigms of vaccine development, and studies on vaccine acceptance rates into its analytical framework. The comprehensive and up-to-date engagement with relevant sources enhanced the study's credibility, providing a well-supported backdrop for its research findings. "I am honored to have the opportunity to read this paper."

Response: Thank you so much for your suggestions and comments. Your suggestion and comments have significantly improved our manuscript. 

At last, we would like to thank respected editor and respected reviewers for the overall positive feedback and help on the improvement of our manuscript and hope that it will be acceptable to them.

Sincerely,

---

## [Editor Report · Decision Letter 2]

29 Feb 2024

Uptake of COVID-19 vaccine among high-risk urban populations in Southern Thailand using the COM-B Model

PONE-D-23-22650R2

Dear Dr. Stanikzai,

We’re pleased to inform you that your manuscript has been judged scientifically suitable for publication and will be formally accepted for publication once it meets all outstanding technical requirements.

Kind regards,

Pei Boon Ooi, Ph.D.

Academic Editor

PLOS ONE
---

## [Editor Report · Acceptance letter]

5 Mar 2024

PONE-D-23-22650R2 

PLOS ONE

Dear Dr. Stanikzai, 

I'm pleased to inform you that your manuscript has been deemed suitable for publication in PLOS ONE. Congratulations! Your manuscript is now being handed over to our production team.

Kind regards, 

on behalf of

Dr. Pei Boon Ooi 

Academic Editor

PLOS ONE